# Untargeted Metabolomics of Slc13a5 Deficiency Reveal Critical Liver–Brain Axis for Lipid Homeostasis

**DOI:** 10.3390/metabo12040351

**Published:** 2022-04-14

**Authors:** Sofia Milosavljevic, Kevin E. Glinton, Xiqi Li, Cláudia Medeiros, Patrick Gillespie, John R. Seavitt, Brett H. Graham, Sarah H. Elsea

**Affiliations:** 1Department of Molecular and Human Genetics, Baylor College of Medicine, Houston, TX 77030, USA; sofia.milosavljevic@bcm.edu (S.M.); kevin.glinton@bcm.edu (K.E.G.); xiqi.li2@bcm.edu (X.L.); john.seavitt@bcm.edu (J.R.S.); 2Harvard Medical School, Boston, MA 02215, USA; 3Department of Medical and Molecular Genetics, Indiana University School of Medicine, Indianapolis, IN 46202, USA; clmedeir@iu.edu (C.M.); pjgilles@iu.edu (P.G.); bregraha@iu.edu (B.H.G.)

**Keywords:** SLC13A5, lipid synthesis, untargeted metabolomics, SLC13A5 deficiency, citrate transport, bile acid metabolism, lipid utilization, liver-brain axis

## Abstract

Though biallelic variants in *SLC13A5* are known to cause severe encephalopathy, the mechanism of this disease is poorly understood. SLC13A5 protein deficiency reduces citrate transport into the cell. Downstream abnormalities in fatty acid synthesis and energy generation have been described, though biochemical signs of these perturbations are inconsistent across SLC13A5 deficiency patients. To investigate SLC13A5-related disorders, we performed untargeted metabolic analyses on the liver, brain, and serum from a *Slc13a5*-deficient mouse model. Metabolomic data were analyzed using the connect-the-dots (CTD) methodology and were compared to plasma and CSF metabolomics from SLC13A5-deficient patients. Mice homozygous for the *Slc13a5^tm1b/tm1b^* null allele had perturbations in fatty acids, bile acids, and energy metabolites in all tissues examined. Further analyses demonstrated that for several of these molecules, the ratio of their relative tissue concentrations differed widely in the knockout mouse, suggesting that deficiency of Slc13a5 impacts the biosynthesis and flux of metabolites between tissues. Similar findings were observed in patient biofluids, indicating altered transport and/or flux of molecules involved in energy, fatty acid, nucleotide, and bile acid metabolism. Deficiency of SLC13A5 likely causes a broader state of metabolic dysregulation than previously recognized, particularly regarding lipid synthesis, storage, and metabolism, supporting SLC13A5 deficiency as a lipid disorder.

## 1. Introduction

Solute carrier family 13 member 5 (SLC13A5) (MIM #608305) is a plasma membrane, sodium-dependent citrate cotransporter and is also referred to as NaCT and INDY [1,2,3]. In this study, we utilized an unbiased analysis of untargeted metabolomic data to investigate the metabolic consequences of citrate transporter deficiency in a *Slc13a5* knockout mouse model and further explored these findings in human data from patient biofluids. Reduced and/or impaired expression and/or function of SLC13A5 homologs in model organisms have been shown to extend lifespan by promoting mitochondrial biogenesis and creating a favorable metabolic state similar to caloric restriction [4,5,6]. Biallelic pathogenic variants in *SLC13A5* result in a severe disorder known as developmental and epileptic encephalopathy 25 with amelogenesis imperfecta (DEE25, MIM #615905). The disorder is characterized by severe pharmaco-resistant seizures, developmental delays, intellectual disability, dental abnormalities, and neonatal encephalopathy [7,8,9,10,11,12,13]. Though symptoms may improve with age, neurological deficits like ataxia, involuntary movements, and spasticity may persist, along with permanent dental abnormalities. Interestingly, elevated expression of the SLC13A5 transporter has been shown in non-alcoholic fatty liver disease (NAFLD).

SLC13A5 contains 12 transmembrane domains and is reported to function as a homodimer to transport citrate and, to a lesser extent, other tricarboxylic acid cycle (TCA) intermediates and dicarboxylic acids [14,15,16]. Due to its role in the transport of citrate, SLC13A5 is implicated as a modulator of fatty acid and cholesterol synthesis, fatty acid oxidation, glycolysis, and gluconeogenesis (Figure 1) [3,17,18]. Citrate is required both as a precursor for de novo fatty acid (FA) synthesis and for the activation of acetyl-CoA carboxylase, the committed step of the pathway [19,20]. SLC13A5 is expressed broadly in the plasma membrane of cells in the brain, testis, kidney, liver, bones, and adrenal gland, with enriched expression in hepatocytes along the sinusoidal membrane [10,19,21,22,23,24].

**Figure 1 metabolites-12-00351-f001:**
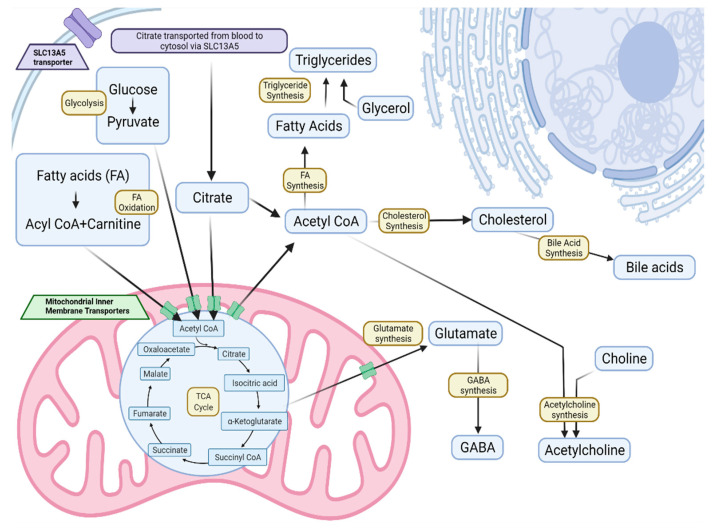
SLC13A5 transport of citrate is critical for lipid synthesis, storage, and utilization. Extracellular citrate is moved into cells where it is converted to acetyl-CoA for lipid synthesis or enters the TCA cycle [27].

In the brain, expression of the transporter has been reported in neurons, astrocytes, and other cell types [2,9,25,26]. Because of these differences and the differing roles of citrate between cell types, deficiency of SLC13A5 may lead to different clinical effects at a tissue level. While processes like fatty acid synthesis, gluconeogenesis, and ketogenesis feature prominently in hepatocytes, neurons depend on citrate to generate energy through the TCA cycle and oxidative phosphorylation [13]. Understanding the nature of these differences in citrate utilization may lead to an increased understanding of the unique effects of SLC13A5 deficiency in different tissues.

The SLC13A5 transporter has been cloned from humans, mice, rats, zebrafish, worms, and fruit flies, though some differences are observed between species [3,28,29,30]. Despite these findings, mouse models have been extensively utilized as models for human SLC13A5-related diseases. Various models with a homozygous knockout of *Slc13a5* in mice show improved tolerance to glucose, a lower percentage of lean body mass, shortening of the tibia, improved memory, heightened grip strength, increased leukocyte count, and abnormal bone and tooth development [3,19,31,32,33,34].

*Slc13a5^−/−^* animals have also recently been shown to exhibit increased neuronal excitability, resulting in a propensity for epileptic seizures ranging from mild myoclonic seizures to generalized tonic-clonic forms [10]. Biochemical analyses of *Slc13a5^−/−^* mice have revealed elevated citrate and malate levels in plasma, with decreased hepatic diacylglycerol levels [35]. Citrate levels have also been shown to be increased in the cerebrospinal fluid (CSF) and decreased in the parahippocampal cortex of *Slc13a5^−/−^* mice, likely reflecting altered transport [10]. However, an explicit mechanism linking abnormal citrate levels to the pathogenesis of seizures has not been clearly described. This gap may be due, in part, to the limited scope of previous biochemical analyses, which have focused primarily on metabolites associated with citrate transport and the TCA cycle. In an effort to uncover novel effects of SLC13A5 deficiency, we performed untargeted metabolomics on the liver, brain, and serum tissue from *Slc13a5^−/−^* mice (*Slc13a5^tm1b/tm1b^*). We are the first to show a subtle endophenotype congruent in both mice and humans via unbiased analyses of untargeted metabolomics data. Further, by employing untargeted metabolomics and comparing differences in metabolite levels between tissues, we were able to map a network of perturbed metabolites which show that the function of this protein extends far beyond citrate transport, as our data point toward abnormalities in bile acids, nucleotide metabolism, and transport and/or synthesis of fatty acids. 

## 2. Results

### 2.1. Tissue-Specific Perturbations in Slc13a5 Deficient Mice

Untargeted metabolomics analyses were conducted on samples of liver, serum, and brain from homozygous (*Slc13a5^−/−^*), heterozygous (*Slc13a5^+/−^),* and wildtype mice. On average, 864 metabolites were identified in serum, 653 in the brain, and 843 in the liver (Appendix A). Connect-the-dots (CTD) analysis was used to determine the molecules and pathways perturbed in *Slc13a5* knockout (KO) animals, as compared to their wildtype (WT) littermates [36]. No sexual dimorphism was observed regarding consistently perturbed metabolites or in CTD outcomes when the analysis was stratified by sex (data not shown). Thus, all data shown include both male and female mice. Metabolomic data from heterozygous *Slc13a5^+/−^* mice were also evaluated; however, metabolic perturbations did not yield meaningful results, suggesting that heterozygosity does not confer statistically significant differences in metabolism (data not shown). Molecules identified as perturbed by CTD comparing *Slc13a5^−/−^* KO mice to WT are presented in the super pathway (Figure 2A) and a sub-pathway (Figure 2B) enrichment analysis by each tissue type. This untargeted analysis identified several common sub-pathways that were perturbed in multiple tissue types. Specifically, fatty acids and derivatives comprised the greatest proportion of perturbed pathways in both liver and brain, and several species were also found perturbed in serum. Bile acids, cholesterols, and sterols proved to be highly perturbed broadly in all tissues examined, comprising the category with either the second or third largest proportion of perturbed metabolites in each tissue. Energy-related sub-pathways were notably perturbed, including nucleotide metabolism (perturbed in all tissues); bile acids, cholesterols, and sterols (all tissues); fatty acids and derivatives (all tissues); monoacylglycerols (liver and brain); carbohydrate and energy metabolism (serum and liver); alanine, aspartate, and glutamate metabolism (serum and brain); sphingolipid metabolism (serum and brain); and glycerophospholipid metabolism (serum and brain) (Figure 2B, Appendix A). 

### 2.2. Tissue Metabolite Perturbations Suggest Abnormal Metabolite Flux in Slc13a5 Deficient Mice

As a plasma membrane transporter, SLC13A5 plays a key role in the transport of molecules into and out of the cell, as well as between tissues. To further explore the tissue-specific perturbations and to examine flux between tissues as a consequence of a Slc13a5 deficiency, the ratios of the batch-normalized, median-scaled levels of metabolites in each tissue were compared within each genotype and then assessed relative to wildtype between each tissue or biofluid (Figure 3A, Appendix A). For example, the relative level of a metabolite in the liver versus the level of that same metabolite in the serum was determined for each genotype, and those ratios were compared: for example, (KO liver/KO serum)*/*(WT liver/WT serum). Ratios were considered altered if <0.5 or >1.5, indicating the fold difference from wildtype (Figure 3A, Appendix A).

For each of the three ratios performed, the following sub-pathways remained significantly abnormal in terms of the ratios between KO and WT in the three tissues: (a) fatty acids and derivatives, (b) monoacylglycerols, (c) glycerophospholipid metabolism, (d) bile acids, cholesterols, and sterols, and (e) sphingolipid metabolism (Figure 3B, Appendix A). Fatty acids, in particular, demonstrate significantly different ratios between tissues when comparing *Slc13a5^−/−^* animals to WT. In *Slc13a5^−/−^* mice, levels of fatty acids were proportionally elevated in the liver when compared to both brain and serum. Fatty acids were also elevated when comparing the brain to serum. These ratios reflect proportionally elevated fatty acid content in the livers of *Slc13a5^−/−^* mice and may suggest a relative paucity of similar species in circulating serum. Interestingly, although fatty acids were generally elevated in the brain compared to the serum, long and very-long-chain fatty acids were decreased. This may suggest an increased breakdown or decreased transport of long (14–22 carbons) and very-long-chain (>22 carbons) fatty acids into the brain tissues of *Slc13a5^−/−^* mice [37].

Though an interpretation of these data at a sub-pathway level is grounding as an overview, specific metabolites within the context of their sub-pathway can be observed and interrogated in the interactive Appendix A at the following link: Slc13a5 Appendix A. Several metabolites with high proportional differences in tissues between *Slc13a5^−/−^* and WT mice were observed, including bilirubin (3.2-fold higher in the liver than in serum) and taurocholate (7.92-fold higher in the brain compared to serum). These changes can be seen due to a perturbed synthesis, degradation, or transport of such molecules and their precursors. The consideration of altered transport of these metabolites becomes more pertinent when considering other metabolites, such as taurocholate, 1-linoleoylglycerol (18:2), nisinate (24:6n3), and 1-oleoylglycerol (18:1). These metabolites are interesting to examine, demonstrating extreme highs in some ratios and extreme lows in others, suggesting that the transport of such molecules is inhibited, resulting in a buildup at the source tissue and scarcity at its destination. In Appendix A, it is also possible to see the total spread of metabolite ratios; whereas, in Figure 3B (Appendix A)*,* only the molecules with the highest and lowest ratios are displayed. Specifically, more subtle differences in molecule flux can be observed when comparing the KO to the WT ratios within the context of their tissues in the review of Appendix A): https://app.powerbi.com/view?r=eyJrIjoiN2I4YTM3ZjMtZGE1MS00ODk0LThlNmItYWIzY2IzZjY2ZGE1IiwidCI6IjZiYmUyYmZmLWNjMzQtNDBlYi05OTZmLWZiNGY2ZjI5MTdiZiIsImMiOjl9.

### 2.3. Biofluid-Specific Perturbations in Patients with Biallelic SLC13A5 Variants

Untargeted metabolomic analyses were previously conducted on samples from five individuals with known SLC13A5 deficiency [15,38]. Previously reported z-scores were collated and assessed to determine the metabolites with the highest and lowest mean z-scores in both plasma (*n* = 8) and cerebrospinal fluid (CSF, *n* = 3) (Table 1, Appendix A). For plasma, two samples were available for each of the three patients, plus one sample each from two patients. CSF samples were single samples, each from a different patient. Similar to the findings in the *Slc13a5^−/−^* mouse model, the most perturbed molecules include fatty acid, bile acid, nucleotide, and energy-related molecules.

In addition to simply analyzing the metabolite z-scores, CTD was employed to better elucidate metabolite perturbations within the context of metabolic pathways and to define the networks of metabolites that are interconnected and perturbed, rather than just considering individual metabolites with obviously high or low z-scores. CTD analysis of the metabolomic profiles from these previously collected plasma and CSF samples uncovered several key perturbed metabolites. Bile acids (e.g., androsterone sulfate, hyocholate, glycohyocholate) and fatty acids (e.g., myristoleate (14:1n5), 17-methylstearate, dodecanedioate) were most perturbed in plasma, while carbohydrate & energy metabolism (e.g., citrate, gluconate) and nucleotides (e.g., N6-succinyladenosine, adenine) were the most perturbed in CSF (Table 2).

### 2.4. Tissue Metabolite Perturbations Suggest Abnormal Metabolite Flux in SLC13A5 Deficient Patients

The levels of individual z-scored metabolites present in both CSF (337 molecules) and plasma (558 molecules) were also assessed within the context of the sample type and the relative degree of perturbation (Figure 4A, Appendix A). While most metabolites, with the exception of citrate, were within a clinically normal range in plasma, several metabolites were altered in CSF. When considering the relative levels of the same metabolite in both biofluids (from a total of 198 metabolites reliably detected and z-scored in both CSF and plasma), 29 metabolites were identified with differential concentrations between plasma and CSF (*p* < 0.05), including carbohydrate and energy metabolites, fatty acids, bile acids, and nucleotides (Figure 4B, Appendix A). These differences in relative levels of metabolites in the two biofluids point toward potential changes in the transport of metabolites in patients with SLC13A5 deficiency. Specifically, altered metabolite transport across the blood-brain barrier can contribute to neurological deviations in patients. Several metabolites, including 4-acetamidobutanoate, inosine, N6-succinyladenosine, and multiple pentose sugars and N-acetylated amino acids, were proportionally elevated in the CSF compared to the plasma from SLC13A5 deficiency patients. Other metabolites were proportionally lower, such as creatine, glucose, glutamate, glycerate, 5-oxoproline, and S-methylcysteine, pointing to sources of altered energy metabolism and impaired management of oxidative stress. Whether these alterations are due to molecule transport or the availability for transport remains to be confirmed. Citrate was elevated in both biofluids and thus, was not differentially altered between compartments.

## 3. Discussion

In our investigation of the metabolomic features of Slc13a5-deficient mice, we both confirmed and broadened our understanding of the importance of this transporter in maintaining normal cellular homeostasis. *Slc13a5^−/−^* mice predictably demonstrated perturbations in fatty acid synthesis and energy metabolism, supporting the idea that Slc13a5 primarily transports citrate, along with other TCA cycle intermediates. We also identified, however, perturbations and compartmental differences in fatty acids and fatty acid derivatives, monoacylglycerols, bile acids, cholesterols, and sterols that add insight into the complexity of lipid and energy metabolism. These data are supported by the work of Pesta et al. (2015) which showed subtle but statistically significant reductions of cholesterol, stearate, and palmitate in fasting plasma from a *Slc13a5* knockdown rat model fed on a high-fat diet [39]. In the liver of this same model, reduced levels of triglycerides and glucose were also reported. These latter reported findings in the liver from high-fat-fed *Slc13a5* knockdown rats are generally consistent with our data that show multiple significantly elevated monoacylglycerols and fatty acids and normal or mildly reduced levels of more complex lipids in *Slc13a5^−/−^* liver, with lower or near-normal levels in serum, compared to wildtype littermates. Taken together, these data further support the need for cytosolic citrate in the synthesis and utilization of both simple and more complex lipids across the organism. One potential caveat is that samples from mice in our study were not collected under fasting conditions, and animals were on normal chow; thus, direct comparisons between these studies are limited.

Studies have shown that impaired citrate transport leads to low cytosolic citrate levels and thus, plausibly decreases the rate of fatty acid and lipid synthesis [40]. This hypothesis is consistent with our findings in the brain, where relative levels of lipids and fatty acids were low. Because neurons rely on circulating fatty acids synthesized elsewhere [41], our study suggests that the underlying etiology may be due to the deficient transport of fatty acids, lipids, and sterols into the brain, perhaps due to previously unknown plasma membrane transport facilitated by Slc13a5, or due to insufficient levels of these molecules available for transport. Similarly, elevated levels of fatty acids and glycerophospholipids within the liver may suggest difficulties with either transporting these species out of the liver (into the blood) or an insufficient synthesis within the cell, limiting the export of certain lipids or due to increased beta-oxidation of some fatty acids or lipids [35]. Increased levels of fatty acids in the liver do not appear to be due to broadly unregulated synthesis, as diacylglycerols and other complex lipids are not observed at increased levels, similar to previously reported studies indicating the absence of fatty liver in other Slc13a5 deficiency mouse models [19,35,39].

We theorize that a deficiency of Slc13a5 compromises citrates transport, resulting in a rate-limiting effect on the cell. Without adequate citrate, cells may only be able to generate sufficient energy to maintain cell metabolism to generate fatty acids, and while the observed increased fatty acids may be due to or compounded by increased beta-oxidation, as previously suggested, the result is that not enough energy is generated to facilitate the synthesis and storage of reserve glycerophospholipids and more complex lipids. These theories could be consistent with the resistance to fatty liver and high-fat diet-induced obesity reported in these animals [19,35]. 

Bile acid perturbations were observed in previous assessments of biofluids from patients with SLC13A5 deficiency; however, at the time, these findings were interpreted as likely due to environmental and dietary confounders [15,38,42]. Data presented here, showing similarly abnormal relative bile acid concentrations in multiple tissues of *Slc13a5^−/−^* mice, confirm the importance of these findings, though the exact pathogenic consequences are unclear. Several studies document the role of bile acids in the brain and their effect on neurological function [43,44,45]. The specific phenotypes of seizures and liver dysfunction that may be associated with bile acid abnormalities warrant the further appraisal of bile acids as the vehicle for the gut-brain connection. 

Sphingolipids are key structural components of mammalian cell membranes, play a crucial role in transport across the cell, maintain membrane flexibility, and are particularly abundant in neurons [46]. Not only do sphingolipids have an immensely important structural role, but they play key roles in complex processes like cell differentiation, cell cycle arrest, apoptosis, and cellular homeostasis. Our finding that sphingolipids are sequestered in the liver, as compared to the brain in the Slc13a5-deficient mice, points toward a potential mechanism for neuronal malfunction if levels and/or composition of sphingolipids are insufficient in the brain.

Glycerophospholipids similarly make up cell membranes, are made in the liver, and play a role in transport across the plasma membrane. Levels of sphingolipids and glycerophospholipids in *Slc13a5^−/−^* mice are proportionally higher in the liver than in the brain, once again suggesting a possible bottleneck in transport. These findings could explain the seizure phenotype strongly present in humans [8,9,11], and just recently, reported in mice [10]. Further, neuronal membrane integrity and altered signaling due to a lack of sphingolipids and glycerophospholipids could be plausible mechanisms to explain ataxia and neurodevelopmental delays in children with SLC13A5 deficiency. Whether these alterations are primarily due to reduced synthesis or transport (or both) remains to be determined; however, targeted intralipids to replace or supplement the reduced lipid content in CSF and the brain may be an opportunity for treatment.

Nucleotide metabolism was perturbed in all three mouse tissues, as identified by the CTD-derived disease module. This finding points to some interesting implications for energy metabolism, generation of ATP, and neurotransmitter metabolism. While the main source of cellular energy for all cells is ATP, neurons rely solely on glycolysis as the energy source. Faulty homeostasis of nucleotide levels in the cell can be detrimental to overall cell health. Tissue-specific neurodegeneration in the brain can occur as a consequence of energy insufficiency, an inability to combat oxidative stress, and impairment of DNA repair [47]. These connections set up a foundation for neurological dysfunction, which is commonly observed in patients with SLC13A5 deficiency [47]. Another concern regarding nucleotide metabolism abnormalities, however, is the potential effect on the synthesis of neurotransmitters, directly impacting neuronal function and creating conditions in which seizures can manifest [47,48,49]. Though the seizure phenotype in both mice and humans needs more study to determine the underlying mechanism, the perturbation of nucleotide metabolism in SLC13A5 deficiency should be more closely examined as a potential contributor to this phenotype.

Although there are inherent differences between mouse and human SLC13A5 function and the effects when deficient, these data support *Slc13a5* knockout animals as accessible and useful model organisms. Specifically, several arguments regarding the difference in phenotypes of the models have been posed, centered around the lack of a seizure phenotype and the presence of liver disease in mice in contrast to the lack of liver disease and presence of seizure phenotype in humans [3]. Although these are legitimate concerns, we believe that further consideration, based on our results and the recent findings of Henke et al. (2021), is warranted [10]. In our analyses, bile acids, glutathione, and glycerophospholipid metabolism were all perturbed in both mice and humans, pointing to previously unrecognized similarities between species regarding perturbations of liver metabolism, gut microbiome, and the brain. Further, Henke et al. presented data supporting abnormal brainwaves in mice, similar to a milder seizure phenotype [10]. Taken together, the Slc13a5 deficiency mouse models could serve as excellent resources for the study of this complex genetic condition. 

Although the primary focus of our study was the discovery of the novel effects of SLC13A5 deficiency, there are many practical lessons that may be applied to the care of these patients. While our data are consistent with the studies of other animal models supporting Slc13a5 deficiency as likely providing protection from fatty liver disease and are in line with previous reports of altered gluconeogenesis and lipid metabolism, additional studies are needed to assess the long-term impact of this altered lipid metabolism and the systemic effects of the metabolic impact on energy metabolism and lipid utilization. This includes the neurological consequences of altered lipid profiles [35,39,50]. Principally, the tenet that these patients are not at risk for liver or kidney disease over their lifespan should be reconsidered, particularly with regard to the use of anti-epileptics, such as valproic acid and other medications known to impact lipid and glucose metabolism [13,46]. Care should be taken to monitor for markers of liver dysfunction, impaired glucose metabolism, and signs of cholestasis—particularly for abnormal levels of bile acids. Furthermore, the consequences of mitochondrial biogenesis as a compensatory mechanism for energy generation across the lifespan have not been investigated. Standards for measuring specific monoacylglycerols, glycerophospholipids, and sphingolipids may be developed and, if implemented, may provide an additional means of disease monitoring and possible drug development. 

Although our analyses specifically point towards perturbations in lipid synthesis, breakdown, or transport in the liver, brain, and blood, more research is required to specifically elucidate the mechanisms of these perturbations. A study tagging citrate and observing its passage through the extra and intracellular environment of the brain, intestines, blood, and brain would be valuable. Understanding the pathway through which SLC13A5 deficiency may increase synthesis, decrease beta-oxidation, or alter the transport of long and very-long-chain fatty acids, could open doors for interventions targeting these abnormalities. Further, additional untargeted metabolomic studies are similarly recommended to confirm our findings; however, we strongly urge against further targeted metabolomics studies focused only on TCA cycle-related metabolites, as we have shown, the impact of SLC13A5 deficiency reaches far beyond this limited scope. Further to this point, Birkenfeld et al. (2011) showed increased expression of *Slc13a3* in *mIndy* knockout mice, suggesting that compensatory overexpression of other citrate transporters may compensate for some aspects of Slc13a5 deficiency; however, *Slc13a3* overexpression was abolished when these mice were fed HFD [35]. Altered expression or function of SLC13A3 has also been associated with a reversible leukoencephalopathy [51], while its temporary ablation in a mouse model of Canavan disease was shown to reduce accumulation of N-acetylaspartate and to improve motor deficits [52]. We still have much to learn regarding the complexities of citrate metabolism and the systemic physiological effects associated with its regulation. 

Finally, we would like to highlight the impact of this study on a broader scale. We are the first to assess untargeted metabolomic data via CTD to precisely define a subtle endophenotype of a genetic disorder. Before our application of this technique, it was thought that the scope of SLC13A5 deficiency was much more limited, simply because only segments of the metabolomic data had been previously examined—and only examined in the context of current knowledge. These data provide further evidence to dispel the notion that targeted metabolomic panels are holistic enough to characterize complex genetic disorders. We advocate for an untargeted approach, because not only does it give light to network-level perturbations that are significantly more complex than a simple biomarker can show, but this approach also gives more clarity on the utility of animal modeling of disease, particularly when the analysis is considered beyond the limits of current knowledge. Considerations in our analyses include the limited numbers of human samples and a lack of repeat analyses; as such, single points in time for any human metabolic analysis can have significant limitations. As with all biochemical analyses, freeze-thaw cycles and overall sample handling could impact the metabolites identified and the relative levels of those metabolites. Further, batch effects in the untargeted metabolomics may impact both human and mouse studies. 

The use and utility of Slc13a5-deficient mouse models have been debated, but as we have shown, the utility of a model system cannot be determined without sufficient exploration of the endophenotype. We hope that our evidence is sufficiently compelling to support the broad implementation of untargeted metabolomic analysis in both humans and mice to fully capture the manifestation of disease. When approaching the analysis of cellular and organismal metabolism solely from a targeted perspective, the data are siloed, limiting the view of the complex, integrated nature of metabolism. Viewing and interpreting these data in this context can be detrimental to the understanding of the disease, impair the use of representative animal modeling, and result in a negative or lack of impact on patient care.

## 4. Methods

### 4.1. Generation of Slc13a5-Deficient Mouse Model, Mouse Husbandry, and Colony Management

All mice studies were reviewed and approved by the Baylor College of Medicine and Indiana University Institutional Animal Care and Use Committees (IACUC) at the respective institutions. Mice harboring a conditional-ready β-galactosidase (lacZ)-tagged reporter allele of *Slc13a5* (C57BL/6N-*Slc13a5^tm1a(EUCOMM)Hmgu^*) were obtained from the European Mouse Mutant Archive repository (https://www.infrafrontier.eu, accessed 23 January 2022) [52,53]. The *Slc13a5^tm1b^* reporter null allele was derived from the *tm1a* allele by Cre excision of the loxP-flanked *Slc13a5* exon 4 (Appendix A). Mice homozygous for the *Slc13a5^tm1b/tm1b^* null allele are viable, with homozygous offspring appearing in litters from heterozygote (Het) pair matings at a frequency of 29% (*n* = 34). The allele does not impair fertility, as homozygote pair matings gave rise to successful pregnancies (https://www.mousephenotype.org/data/genes/MGI:3037150, accessed 23 January 2022).

Mice heterozygous for the *Slc13a5* null allele (*Slc13a5^+/tm1b^*) were maintained on the C57BL/6N background strain and genotyped using polymerase chain reaction (PCR) using a three-oligonucleotide primer strategy consisting of allele-specific forward primers (F1 for the wildtype allele and F2 for *Slc13a5^tm1b^* allele) combined with a single reverse primer (R) common to both alleles (Appendix A). The wildtype PCR amplicon (F1 + R) is 919 bp, and the *Slc13a5^tm1b^* amplicon (F2 + R) is 1301 bp. F1 = AGAGATTATAACTGGGCTCACGA; F2 = CGTCAGTATCGGCGGAATTC; R = TCTTCGTCTTCCTGCTCCTG. *Slc13a5^tm1b/tm1b^* null allele mice are represented as *Slc13a5^−/−^* or KO in this study.

Male and female mice were analyzed in this study for each tissue type: serum (WT, *n* = 10, male (m) = 7, female (f) = 3; Het, *n* = 11, m = 6, f = 5; KO, *n* = 9, m = 3, f = 6), liver (WT, *n* = 10, m = 6, f = 4; Het, *n* = 12, m = 7, f = 5; KO, *n* = 10, m = 4, f = 6), or brain (WT, *n* = 10, m = 6, f = 4; Het, *n* = 12, m = 7, f = 5, KO, *n* = 9, m = 4, f = 5). Males and females were group-housed, 2–5 mice per microisolator cage, in ventilated racks under controlled temperature of 22–24 °C and 50–60% humidity, in a 12:12 h light/dark cycle. Mice had access to autoclaved water and food ad libitum (Teklad 2018SX). Standardized phenotyping (https://www.mousephenotype.org/impress, accessed 23 January 2022) was completed through the International Mouse Phenotyping Consortium, with primary data and analysis available at (https://www.mousephenotype.org/data/genes/MGI:3037150, accessed 23 January 2022).

### 4.2. Mouse Tissue Sample Preparation

Male and female animals were euthanized between 54–63 days at ~8–9 weeks of age, and whole brain, whole liver, and serum samples were collected. Samples were promptly flash-frozen in liquid nitrogen and subsequently stored at –80 °C until processed. The sample preparation was carried out as described previously at Metabolon, Inc (Morrisville, NC, USA) [54,55]. Briefly, samples were prepared using the automated MicroLab STAR^®^ system (Hamilton Company, Franklin, MA, USA). Several recovery standards were added prior to the first step in the extraction process for quality control purposes. To remove protein and to recover chemically diverse metabolites, small molecules bound to protein or trapped in the precipitated protein matrix were dissociated with methanol under vigorous shaking for 2 min (Glen Mills GenoGrinder 2000) and precipitated by centrifugation. The resulting extract was divided into five fractions: two for analysis by two separate reverse-phase (RP)/UPLC-MS/MS methods with positive ion mode electrospray ionization (ESI), one for analysis by RP/UPLC-MS/MS with negative ion mode ESI, one for analysis by HILIC/UPLC-MS/MS with negative ion mode ESI, and one sample was reserved for backup. Samples were placed briefly on a TurboVap^®^(Zymark, Hopkinton, MA, USA) to remove the organic solvent. The sample extracts were stored overnight under nitrogen before preparation for the analysis. 

### 4.3. Human Plasma and CSF Sample Collection and Analysis

Previously reported clinical untargeted metabolomic profiling data obtained from individuals with a SLC13A5 deficiency were utilized in this study [15,38,55,56]. Data reported in the original study by Bainbridge et al. (2017) and in clinical testing cohorts [38] were reanalyzed, as described herein [15]. The collated untargeted metabolomics human dataset was independently analyzed within the context of human metabolite alterations and also analyzed in comparison to *Slc13a5^−/−^* mouse metabolite perturbations. The untargeted metabolomic profiles generated from the biofluids of individuals with SLC13A5 deficiency used in this study were: plasma (*n* = 8 samples from 5 patients; 3 male, 2 female) and CSF (*n* = 3 samples from 3 patients; 2 male, 1 female). See Appendix A. 

### 4.4. Untargeted Metabolomics

Metabolomic analyses were conducted by Metabolon, Inc. using ultra-high-performance liquid chromatography/tandem mass spectrometry (UHPLC/MS/MS), as previously described [57]. Briefly, samples were analyzed through four different methods: (1) acidic positive ion conditions, chromatographically optimized for more hydrophilic compounds, (2) acidic positive ion conditions optimized for more hydrophobic compounds, (3) basic negative ion optimized conditions using a separate dedicated C18 column, and (4) negative ionization following elution from a HILIC column. All methods utilized a Waters ACQUITY ultra-performance liquid chromatography (UPLC) and a Thermo Scientific Q-Exactive high resolution/accurate mass spectrometer interfaced with a heated electrospray ionization (HESI-II) source and Orbitrap mass analyzer, operated at 35,000 mass resolution. The MS analysis alternated between MS and data-dependent MS^n^ scans using dynamic exclusion. The scan range varied slighted between methods but covered 70-1000 m/z. Human metabolomic profiles were generated using the same methods, except using a CLIA-CAP-validated clinical platform, as previously described for *n* of 1 clinical metabolomics [38,42,55,56]. All human data assessed were provided as z-scores derived relative to the control reference population for each sample type (plasma or CSF). Scaled data provided by Metabolon, Inc. were utilized for mouse tissue analyses. Within each tissue type, original scaled data for each metabolite were rescaled so that the median value for that metabolite in all the samples was set to one, and all other samples were scaled relative to this value. For samples in which a particular metabolite was either missing or not detected, the value was imputed with the scaled minimum value. Metabolites that failed detection in at least 50% of either male or female samples were removed from analysis in that particular tissue type. This resulted in the removal of approximately 10% of metabolites for each tissue type (9% serum, 11% brain, and 9% liver). 

### 4.5. Metabolomics Data Analysis 

To interpret the untargeted metabolomics data, we employed the previously reported CTD method, or the “connect the dots” information-theoretic algorithm that was designed to efficiently interpret metabolic perturbations in the context of network models [36]. Briefly, the method takes a weighted network and a set of network nodes as inputs and identifies a subset of the input nodes that are highly connected within the network. The method also provides a *p*-value corresponding to the level of connectedness of the input node subset within the network. These network models contain metabolite nodes, where edges connect metabolites that are co-perturbed in a set of cases in contrast to controls. Gaussian graphical models are used to compute edge weights, which indicate the strength of the positive or negative partial correlation between metabolites. CTD was employed to create a network that encompassed all of the cases compared to the controls (disease module CTD), in addition to each case individually compared to the controls (case-specific CTD). The network model is learned to represent the disease/condition in two steps: First, two types of graphical models were constructed, one containing both Slc13a5-deficient and control samples (case + control network) and a second from only control samples (control network). Next, the edges found in the case + control network that were also found in the control network were pruned. We then applied CTD to calculate a “Slc13a5 deficiency model” in the pruned, Slc13a5 deficiency-specific network. The Student’s *t*-test, *p* < 0.05, was considered significant.

For the human metabolomics data, CTD was employed to create networks both for plasma and CSF, as detailed above. Human population reference controls were used [36,58]. By employing the case-specific CTD, we were able to capture more metabolites and obtain a count of the number of times a metabolite showed up in a participant’s CTD. The relative levels of metabolites present in both CSF and plasma were also assessed to consider significant differences between the compartments in SLC13A5 deficiency compared to clinical reference populations. The Student’s *t*-test, *p* < 0.05, was considered significant.

## Figures and Tables

**Figure 2 metabolites-12-00351-f002:**
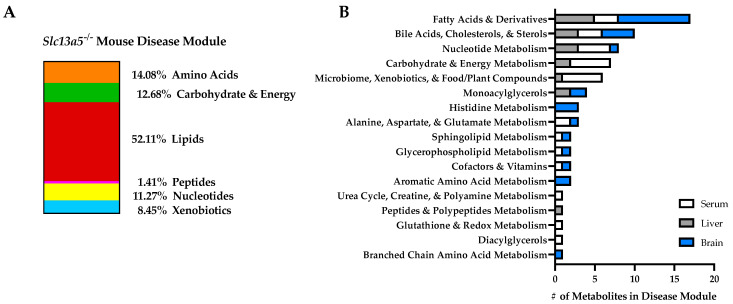
Slc13a5 deficiency mouse model disease modules determined by CTD. CTD analysis identified the most perturbed metabolites in tissues from *Slc13a5^−/−^* mice as compared to wildtype mice for each tissue tested: serum (WT, *n* = 10; KO, *n* = 9), liver (WT, *n* = 10; KO, *n* = 10), or brain (WT, *n* = 10; KO, *n* = 9). Pathways within which perturbed metabolite falls are shown to provide a more meaningful analysis than simply listing individual metabolites. (**A**) CTD analysis represented in an enrichment of the super pathways for the key metabolites in the CTD network. Pathway enrichment of the CTD networks for the 71 CTD-identified metabolites identified in the brain (*n* = 26), serum (*n* = 28), and liver (*n* = 17) are illustrated in the major metabolic super pathways, highlighting the >50% lipid composition of these disease modules. (**B**) Sub-pathway enrichment of the CTD network of the most perturbed metabolites and pathways in Slc13a5 deficiency mice are shown for serum, brain, and liver (for tissue metabolite detail, see Appendix A). # = number or quantity of metabolites.

**Figure 3 metabolites-12-00351-f003:**
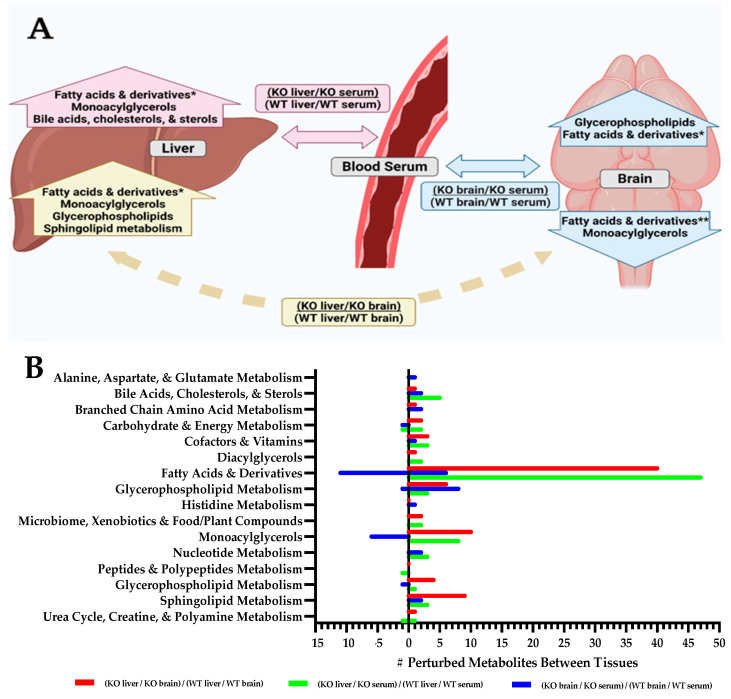
Metabolite flux between tissues is altered in *Slc13a5* deficiency mice. (**A**) A model of the flow of metabolites between tissues in *Slc13a5^−/−^* mice is shown. Horizontal arrows indicate the metabolite ratios assessed to understand the change in flux, storage, synthesis, or breakdown of metabolites between *Slc13a5^−/−^* and WT mice. Mice included for each tissue type: serum (WT, *n* = 10; KO, *n* = 9), liver (WT, *n* = 10; KO, *n* = 10), or brain (WT, *n* = 10; KO, *n* = 9). Arrows pointing vertically represent types of metabolites whose proportional levels change between tissues in *Slc13a5^−/−^* mice compared to WT mice. * = mix of short, medium, long, and very-long-chain fatty acids ** = only contain long (14–22 carbons) and very-long-chain (>22 carbons) fatty acids [27]. See Appendix A. (**B**) Perturbed metabolite ratios between tissues and genotypes illustrating altered distribution. The numbers of molecules (#) in sub-pathways with ratios of <0.5 (left side of *x*-axis) and ratios of >1.5 (right side of *x*-axis) are represented. These altered ratios indicate that the proportionality of the metabolite between the two tissues of comparison is perturbed in the citrate transporter KO condition. The left side of the *x*-axis shows the numbers of molecules (#) in each sub-pathway with ratios of <0.5, indicating that in the KO, the metabolite is proportionally low in the numerator tissue. The right side of the *x*-axis shows the number of molecules (#) in each sub-pathway that had altered ratios between tissues that were >1.5 in the KO, indicating a higher proportional quantity of the metabolite in the numerator tissue compared to WT littermates. (see Appendix A). Wildtype = WT; Knockout = KO. Detailed analysis of metabolites can be viewed in the interactive Appendix A.

**Figure 4 metabolites-12-00351-f004:**
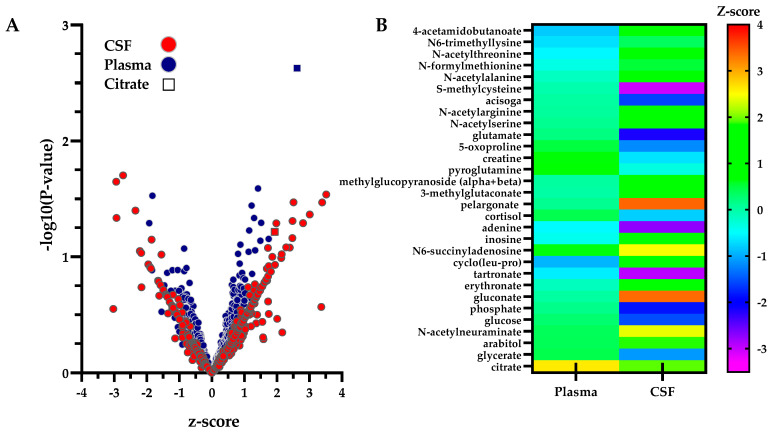
Comparison of Metabolites in Human Biofluids Illustrates Significant Alterations Between Plasma and CSF. (**A**) Metabolomic profiles of SLC13A5 deficiency in plasma and CSF are shown. Metabolites present in plasma (*n* = 8, blue-filled circles) and cerebrospinal fluid (CSF) (*n* = 3, red-filled circles) obtained from individuals with SLC13A5 deficiency (see Methods) are illustrated in a volcano plot. Citrate is indicated as a square for each biofluid for ease of identification. (**B**) Compartmental differences between plasma and CSF suggest altered metabolic flux. Molecules significantly altered in CSF compared to plasma are illustrated in a heat map (*p* < 0.05 Student’s *t*-test) to highlight compartmental differences in the flux of these molecules. Citrate, elevated in both biofluids, is included in the heatmap; however, citrate flux is not significantly different between plasma and CSF. Molecules not identified in both biofluids were not assessed for flux. See Appendix A.

**Table 1 metabolites-12-00351-t001:** Significantly Altered Metabolites in Plasma and Cerebrospinal Fluid in Patients with SLC13A5 Deficiency.

CSF	Biochemical	Sub-Pathway	Mean z-Score * ±S.D.
High	pelargonate (9:0)	Fatty Acids & Derivatives	3.4 ± 0.4
	gluconate	Carbohydrate & Energy Metabolism	3.4 ± 4.4
	N-acetylglutamine	Alanine, Aspartate, & Glutamate Metabolism	2.5 ± 0.7
	N6-succinyladenosine	Nucleotide Metabolism	2.5 ± 1.0
	N-acetylneuraminate	Carbohydrate & Energy Metabolism	2.4 ± 1.4
	citrate	Carbohydrate & Energy Metabolism	1.9 ± 0.4
Low	glutamate	Alanine, Aspartate, & Glutamate Metabolism	−2.2 ± 2.0
	4-guanidinobutanoate	Urea Cycle, Creatine, & Polyamine Metabolism	−2.4 ± 0.6
	adenine	Nucleotide Metabolism	−2.7 ± 0.4
	tartronate (hydroxymalonate)	Microbiome, Xenobiotics, & Food/Plant Compounds	−2.9 ± 0.7
	S-methylcysteine	Sulfur Containing Amino Acid Metabolism	−3.0 ± 4.0
**Plasma**	**Biochemical**	**Sub-Pathway**	**Mean z−Score * ± S.D.**
High	citrate	Carbohydrate & Energy Metabolism	2.6 ± 0.8
	malonate	Fatty Acids & Derivatives	1.7 ± 1.1
	1-arachidonoylglyercophosphate	Glycerophospholipid Metabolism	1.5 ± 0.9
	pentadecanoate (15:0)	Fatty Acids & Derivatives	1.5 ± 0.9
	docosapentaenoate (n6 DPA; 22:5n6)	Fatty Acids & Derivatives	1.4 ± 0.6
	glycocholate sulfate *	Bile Acids, Cholesterols, & Sterols	−1.3 ± 1.4
Low	5alpha-pregnan-3beta,20alpha-diol disulfate	Bile Acids, Cholesterols, & Sterols	−1.5 ± 1.5
	5alpha-androstan-3beta,17beta-diol disulfate	Bile Acids, Cholesterols, & Sterols	−1.5 ± 2.3
	1-palmitoylglycerol (16:0)	Monoacylglycerols	−1.8 ± 1.0
	suberate (octanedioate)	Fatty Acids & Derivatives	−1.9 ± 0.2

* Unlike the mouse data described in this study, for human samples, a clinical reference population was utilized to generate z-scores relative to the control reference population for each metabolite for analysis of *n* of 1 clinical sample. Some metabolites had particularly high or low z-scores, highlighting these metabolites as associated with the SLC13A5 deficiency metabolic phenotype (see details in Appendix A). Metabolomic profiling was previously reported in Bainbridge et al. (2017) and Alaimo et al. (2020). Standard deviation = S.D.; Cerebrospinal fluid = CSF.

**Table 2 metabolites-12-00351-t002:** Metabolites in the SLC13A5 Deficiency Disease Modules in Plasma and Cerebrospinal Fluid from Patients with SLC13A5 Deficiency as Determined by CTD.

Biochemical	Plasma	
	Super Pathway	Sub-Pathway
androsterone sulfate	Lipids	Bile Acids, Cholesterols, & Sterols
epiandrosterone sulfate	Lipids	Bile Acids, Cholesterols, & Sterols
5alpha-pregnan-3beta,20alpha-diol-disulfate	Lipids	Bile Acids, Cholesterols, & Sterols
hyocholate	Lipids	Bile Acids, Cholesterols, & Sterols
glycohyocholate	Lipids	Bile Acids, Cholesterols, & Sterols
citrate	Carbohydrate & Energy Metabolism	Carbohydrate & Energy Metabolism
adipate	Lipids	Fatty Acids & Derivatives
myristoleate (14:1n5)	Lipids	Fatty Acids & Derivatives
methylpalmitate (15 or 2)	Lipids	Fatty Acids & Derivatives
mead acid (20:3n9)	Lipids	Fatty Acids & Derivatives
docosapentaenoate (n6 DPA; 22:5n6)	Lipids	Fatty Acids & Derivatives
17-methylstearate	Lipids	Fatty Acids & Derivatives
dodecanedioate	Lipids	Fatty Acids & Derivatives
2-oleoyl-GPE (18:1)	Lipids	Glycerophospholipid Metabolism
3-methylxanthine	Xenobiotics	Microbiome, Xenobiotics & Food/Plant Compounds

**Biochemical**		**CSF**
	**Super Pathway**	**Sub-Pathway**
N-acetylglutamine	Amino Acids	Alanine, Aspartate, & Glutamate Metabolism
N-acetylalanine	Amino Acids	Alanine, Aspartate, & Glutamate Metabolism
N-acetylneuraminate	Carbohydrate & Energy Metabolism	Carbohydrate & Energy Metabolism
citrate	Carbohydrate & Energy Metabolism	Carbohydrate & Energy Metabolism
2-methylcitrate	Carbohydrate & Energy Metabolism	Carbohydrate & Energy Metabolism
erythronate	Carbohydrate & Energy Metabolism	Carbohydrate & Energy Metabolism
gluconate	Carbohydrate & Energy Metabolism	Carbohydrate & Energy Metabolism
ribitol	Carbohydrate & Energy Metabolism	Carbohydrate & Energy Metabolism
ribonate	Carbohydrate & Energy Metabolism	Carbohydrate & Energy Metabolism
pyridoxate	Carbohydrate & Energy Metabolism	Cofactors & Vitamins
pyridoxal	Carbohydrate & Energy Metabolism	Cofactors & Vitamins
malonylcarnitine	Lipids	Fatty Acids & Derivatives
tartronate (hydroxymalonate)	Xenobiotics	Microbiome, Xenobiotics, & Food/Plant Compounds
N6-succinyladenosine	Nucleotide Metabolism	Nucleotide Metabolism
adenine	Nucleotide Metabolism	Nucleotide Metabolism
gamma-glutamylthreonine	Peptides	Peptides & Polypeptides Metabolism
S-methylcysteine	Amino Acids	Sulfur Containing Amino Acid Metabolism
acisoga	Amino Acids	Urea Cycle, Creatine, & Polyamine Metabolism

Cerebrospinal fluid = CSF. Note: All key molecules identified in the CTD network analysis are shown. These data highlight the overlap between mouse (Figure 2B) and human data for bile acids, cholesterols, and sterols, nucleotide metabolism, and carbohydrate and energy metabolism. Data represent human samples: CSF (2 male, 1 female) and plasma (5 male, 3 female).

## Data Availability

The data presented in this study are available in the article, online eTable link, and in the Appendix A.

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
