# Peer review of "Untargeted Metabolomics of Slc13a5 Deficiency Reveal Critical Liver–Brain Axis for Lipid Homeostasis"

_metabolites, 2022, doi:10.3390/metabo12040351_

Round 1
Reviewer 1 Report
I find your work interesting, excellently described and discussed and believe that your manuscript deserve to be publish in present form. However, in your manuscript I find just two omissions, technical inconsistencies which may be possible consequence of the journal editorial policy:
In Results section, Par. 4, Line 201: It is not possible to follow the link: Slc13a5 eTable 1, and thereby observed and interrogated specific metabolites within the context of their sub-pathway in the interactive eTable 1 !?; and in
Supplementary Materialses stement, Par. 1, Line 530: The supporting information can not be downloadad at provided hyperlink (www.mdpi.com/xxx/s1) – (Error 404 - File not found) !?
I kindly paying attention to those two details which I am sure contribute to the clarity of the text and improve the entire manuscript.
Author Response
Thank you very much for taking the time to review our manuscript and for your suggested revisions. We have attached your comments and our responses. Your feedback was very helpful and has improved the clarity of the paper.
1.In Results section, Par. 4, Line 201: It is not possible to follow the link: Slc13a5 eTable 1, and thereby observed and interrogated specific metabolites within the context of their sub-pathway in the interactive eTable 1 !?
Response: We are not sure why the direct link did not work for you, but we will provide the full URL to Metabolites to ensure the link can be followed. Here is the full link: https://app.powerbi.com/view?r=eyJrIjoiN2I4YTM3ZjMtZGE1MS00ODk0LThlNmItYWIzY2IzZjY2ZGE1IiwidCI6IjZiYmUyYmZmLWNjMzQtNDBlYi05OTZmLWZiNGY2ZjI5MTdiZiIsImMiOjl9
- Supplementary Materialses stement, Par. 1, Line 530: The supporting information can not be downloadad at provided hyperlink (www.mdpi.com/xxx/s1) – (Error 404 - File not found) !?
Response: We think this was an error in the version that was sent to you. Our apologies, but we will clarify the addition and address of the link with Metabolites, as the Supplemental Materials should have been accessible to you.
Reviewer 2 Report
Milosavljevic and colleagues propose a clear characterisation of the metabolic profile of the Slc13a5 deficient mouse model, and the analysis of SLC13A5 deficient patients biofluids is an added value. Overall the manuscript is well written and concepts are clearly stated. However I have some observations:
- I would suggest to explain better the aim of the study in the introduction, and it would be more clear if the aim is clearly stated in the firsts sentences of the introduction and not after all the explanation of the Slc13a5 transporter
- numbers of Male and female animals should be better clarified, how many females ? males? Have you observed any gender differences ? Please discuss this point.
- plasma analysis on human patients were performed on both sexes? This is not specified in the method section, I suggest to add this information
- I would also suggest to add a part of discussion about the future perspectives of this work
Author Response
Thank you very much for taking the time to review our manuscript and for your suggested revisions. We provided responses to your comments and our responses below. Your feedback was very helpful and has improved the structure and clarity of the paper.
- I would suggest to explain better the aim of the study in the introduction, and it would be more clear if the aim is clearly stated in the firsts sentences of the introduction and not after all the explanation of the Slc13a5 transporter
Response: This change has been made as requested, and now the paper is hopefully clearer from the start. Thank you for the suggestion.
- numbers of Male and female animals should be better clarified, how many females ? males? Have you observed any gender differences? Please discuss this point.
Response: Thank you for pointing out this omission. We have added the number of male and female mice in Section 4. Methods. We have also highlighted our descriptions of the lack of sex differences we found in Section 2.1 Tissue specific perturbations in Slc13a5 deficient mice. Please let us know if further discussion is needed, and we would be happy to elaborate.
- plasma analysis on human patients were performed on both sexes? This is not specified in the method section, I suggest to add this information
Response: This is a good point, and we thank you for identifying this omission on our part. We have added the numbers of male and female patients in the human data.
- I would also suggest to add a part of discussion about the future perspectives of this work
Response: We have addressed avenues in which to continue this work in lines 383-394:
“Though our analyses specifically point towards perturbations in lipid synthesis, breakdown, or transport in the liver, brain, and blood, more research is required to specifically elucidate the mechanisms of these perturbations. A study tagging citrate and observing its passage through the extra and intra-cellular environment of the brain, intestines, blood, and brain would be valuable, as well as additional studies to track lipid processing and transport in this condition. Studies to target the pathway(s) through which SLC13A5 deficiency increases synthesis, decreases beta-oxidation, or alters transport of long and very long chain fatty acids could open doors for interventions targeting these abnormalities. Further, additional untargeted metabolomic studies in individuals with SLC13A5 deficiency are needed to confirm our findings; however, we strongly urge against further targeted metabolomic studies focused only on TCA cycle-related metabolites, as we have shown that the impact of SLC13A5 deficiency reaches far beyond this limited scope.”
Please let us know if further elaboration is needed.
Reviewer 3 Report
In this interesting work, the authors have conducted an untargeted metabolomic analysis on metabolites in liver, brain and serum from a SLC13A5 deficient mouse model. They performed connect-the-dots analysis and compare the results to those from human patients. They have shown that SLC13A5 knockdown resulted in "broad spectrum" effects and make the case for carrying out untargeted analysis instead of focusing primarily on specific pathways such as TCA cycle and/or other pathways connected to central metabolism.
Some comments:
(1) Please include the detailed description of the model used in CTD analyses. Since there are more than 800 metabolites in the analyses, the networks themselves would be very interesting
(2) How did the data differ between male and female mice? How many mice of each sex was used? The authors mention data stratified by sex was not significant but do not mention how many mice were used or show the data.
(3) Please include a section on limitations of this analysis. The authors make a good case for using untargeted analysis. However, some limitations should be discussed.
Author Response
Thank you very much for taking the time to review our manuscript and for your suggested revisions. We have provided your comments and our responses below. Your feedback was very helpful and has improved the structure and clarity of the paper.
(1) Please include the detailed description of the model used in CTD analyses. Since there are more than 800 metabolites in the analyses, the networks themselves would be very interesting
Response: Thank you for pointing out this lack of clarity on our part. We have made revisions that we hope will make the CTD results presented in this study clearer to the reader. As the CTD data provided in the paper was comprehensive, our presentation of the data should have been more clear. Please see our enhanced descriptions for Figure 2 (mouse CTD information/data) and Table 2, which details the human CTD. Specifically, with regard to Figure 2, we contemplated showing each metabolite in each network but found that there was more meaning to the data when the metabolites identified in CTD analysis were represented by the molecule super pathway (Figure 2A) and sub-pathway (Figure 2b). We hope the clarifications provided improve this understanding. Please let us know if you have other questions or recommendations.
(2) How did the data differ between male and female mice? How many mice of each sex was used? The authors mention data stratified by sex was not significant but do not mention how many mice were used or show the data.
Response: Thank you for pointing out our omission. We have added the number of male and female mice in Section 4 Methods. We have also highlighted our descriptions of the lack of sex differences in our data in Section 2.1 Tissue specific perturbations in Slc13a5 deficient mice. Please let us know if further discussion is needed, and we would be happy to elaborate.
(3) Please include a section on limitations of this analysis. The authors make a good case for using untargeted analysis. However, some limitations should be discussed.
Response: We have added a discussion on the limitations of our study in the Discussion section, thank you for this feedback.